# Phone-based psychosocial counseling for people living with HIV: Feasibility, acceptability and impact on uptake of psychosocial counseling services in Malawi

## Research Article

mental health; HIV; Africa; tele-health

**Corresponding author:**
Carrie M. Cox;
Email: cc12@bcm.edu

Carrie M. Cox[1,2] , Steven Masiano[1], Alick Mazenga[1], Madeline Stark[1],
Michael Udedi[3], Katherine R. Simon[1,2], Saeed Ahmed[1,2], Phoebe Nyasulu[1] and
Maria H. Kim[1,2]

[1]Baylor College of Medicine Children's Foundation Malawi, Lilongwe, Malawi; [2]Baylor College of Medicine, Texas Children's Hospital, Houston, USA and [3]Curative, Medical and Rehabilitation Services – Mental Health, Malawi Ministry of Health, Lilongwe, Malawi

## Abstract

People living with HIV experience psychosocial needs that often are not addressed. We designed an innovative low-resource model of phone-based psychosocial counseling (P-PSC). We describe cohort characteristics, acceptability, feasibility and utilization of P-PSC at health facilities supported by Baylor Foundation Malawi. Staff were virtually oriented at 120 sites concurrently. From facility-based phones, people with new HIV diagnosis, high viral load, treatment interruption or mental health concerns were referred without identifiable personal information to 13 psychosocial counselors via a WhatsApp group. Routine program data were retrospectively analyzed using univariate approaches and regressions with interrupted time series analyses. Clients utilizing P-PSC were 63% female, 25% youth (10–24 y) and 9% children (<10 y). They were referred from all 120 supported health facilities. Main referral reasons included new HIV diagnosis (32%), ART adherence support (32%) and treatment interruption (21%). Counseling was completed for 99% of referrals. Counseling sessions per month per psychosocial counselor increased from 77 before P-PSC to 216 in month 1 (95% CI = 82, 350, p = 0.003). Total encounters increased significantly to 31,642 in year 1 from ~6,000 during the 12 prior months, an over fivefold increase. P-PSC implementation at 120 remote facilities was acceptable and feasible with immediate, increased utilization despite few psychosocial counselors in Malawi.

## Impact statement

People living with HIV (PLHIV) frequently need psychosocial health services, but access remains suboptimal. The shortage of skilled mental healthcare providers is particularly acute in sub-Saharan Africa, home to most PLHIV. In Malawi, lay community health workers provide most first-line facility-based counseling for PLHIV and refer those clients who have needs that exceed their counseling capacity to psychosocial counselors, a cadre more extensively trained in and dedicated to counseling. The Baylor Foundation Malawi's Tingathe program has been supporting psychosocial counselors at 10 high-volume facilities to provide in-person counseling since 2017; and in 2020, to continue service delivery amidst COVID-19 mitigation strategies, phone-based psychosocial counseling (P-PSC) services were started. We describe this P-PSC service's implementation, feasibility and acceptability in Malawi. A pool of 13 psychosocial counselors provided phone-based counseling to clients from 120 health facilities. Our results show that P-PSC was acceptable and feasible in Malawi. P-PSC expanded access to psychosocial counselors twelvefold, from 10 to all 120 supported health facilities, and over 30,000 counseling sessions were conducted in the first year alone. P-PSC was delivered simultaneously to many geographically dispersed facilities with a core team of trained providers. P-PSC has shown great promise as an efficient and scalable model of counseling by trained psychosocial counselors in limited resource settings with shortages of skilled mental health workers.

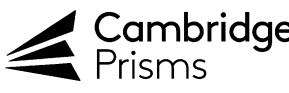



## Background

People living with HIV (PLHIV) commonly experience psychosocial health needs, including challenges coping with illness, stigma and discrimination, intimate partner violence, depression, adherence and retention (Li et al., 2014; Rueda et al., 2016; Tao et al., 2018; Remien et al., 2019; Dessauvagie et al., 2020). Psychosocial counseling can help people improve their well-being and

overcome barriers to retention and adherence to HIV care, allowing them to achieve viral load suppression and ultimately improve survival (Fox and Rosen, 2015; Geldsetzer et al., 2016; Casale et al., 2019; WHO, 2021).

In sub-Saharan Africa where over 70% of PLHIV (37.7 million) reside, access to psychosocial support services is very limited (UNAIDS, 2020). There are less than 1.4 estimated mental health workers per 100,000 people in the African region compared to the global average of 9.0 per 100,000 (WHO, 2017). In Malawi, the public government-supported health system provides care for most of the 18 million people with only two psychiatrists on its staff (Chorwe-Sungani, 2021). Additionally, gaps in awareness and pervasive stigma surrounding mental illness and psychological distress likely contribute to poor access to mental health services (Crabb et al., 2012; WHO, 2017; Adepoju, 2020; Galagali and Brooks, 2020). Some promising interventions that work with adolescents living with HIV and pregnant women aim to address these gaps in the delivery of mental health services. They achieve this by training lay cadres to provide in-person and virtual counseling, as well as by creating text messaging-based support group interventions (Chibanda et al., 2016; Skinner et al., 2018; Atujuna et al., 2021; Dambi et al., 2022; Bengtson et al., 2023). Critical work exploring the complexities and nuance of adapting such interventions to a larger scale is ongoing (Chibanda, 2017; Stockton et al., 2020; Moitra et al., 2023).

With a long history of task shifting and engaging lay cadres in basic counseling, particularly in HIV care and treatment services, the Malawi government addressed the shortage of skilled mental health providers by introducing "psychosocial counselors" to provide counseling services (Flick et al., 2019). Psychosocial counselors undergo a two-year postsecondary school training program that includes didactics and a supervised clinical internship to develop the skills necessary to address complex psychosocial needs. These psychosocial counselors work with clients who have complex needs and are referred to them by lay providers who deliver basic counseling services at primary health facilities across Malawi.

To support the Malawi government in providing quality psychosocial counseling services, the USAID/PEPFAR-supported program Tingathe (meaning "together we can") hired and trained more facility-based psychosocial counselors. Tingathe is a Baylor Foundation Malawi program in partnership with the Ministry of Health that supports HIV care and treatment services (Kim et al., 2012; Kim et al., 2013; Ahmed et al., 2015). Psychosocial counselors are among the lay and professional healthcare workers (HCWs) who provide free HIV testing, care and treatment services alongside Ministry of Health colleagues at supported facilities. At the district and central levels, Tingathe staff provide additional in-person and remote technical support, including supervision and evaluation of interventions for health system strengthening and quality improvement (Kim et al., 2012; 2015). In 2020, with the COVID-19 pandemic disrupting the provision of in-person psychosocial counseling, Baylor Foundation Malawi pivoted to and accelerated the delivery of psychosocial counseling services over the phone – a form of telehealth.

Telehealth, or telemedicine, is exchanging medical information using electronic communication to improve a patient's health (Tuckson et al., 2017). It provides an opportunity to improve access to psychosocial services and extend the reach of a limited pool of expert providers (Adeloye et al., 2017; Ag Ahmed et al., 2017; Naslund et al., 2017; Kaonga and Morgan, 2019). In limited resource settings, telehealth has been used to provide patient consultation, provider education, screening for mental illness, adherence support and self-help (Naslund et al., 2017; Clough et al., 2019; Kaonga and Morgan, 2019). With increasing mobile phone penetration in limited resource settings, more opportunities for telehealth have emerged.

The expansion of psychosocial counseling coverage through telehealth services had been considered in Malawi, but was not previously implemented due to lack of experience, concerns over limited mobile phone penetration, as well as the ongoing stigma surrounding mental illness (Crabb et al., 2012; Udedi, 2016; Handforth and Wilson, 2019). In 2020, due to COVID-19, increasing safety by decongesting health facilities and minimizing movement became a national priority. As such, existing facility-based support services became more limited. Therefore, despite previously identified potential barriers, telehealth psychosocial counseling was implemented because it became one of the only means to ensure continued service delivery. Tingathe has built on the telehealth experience in other limited resource settings and implemented phone-based psychosocial counseling (P-PSC) to allow continued service delivery amidst COVID-19 mitigation restrictions.

We describe the acceptability and feasibility of P-PSC in Malawi and evaluate the impact of P-PSC on the geographic coverage and utilization of psychosocial counseling among PLHIV at health facilities supported by Baylor Foundation Malawi.

## Methods

This was a retrospective analysis of routinely collected program data using a single-group interrupted times series design. The Malawi National Health Sciences Research Committee and Baylor College of Medicine Institutional Review Board approved this study.

## Setting

The P-PSC intervention was implemented at 120 public and private health facilities supported by Tingathe within seven districts in central and southeast Malawi, namely Balaka, Lilongwe, Machinga, Mangochi, Mulanje, Phalombe and Salima. All facilities in Malawi provide free HIV care and treatment services. The combined catchment population of the facilities was 4.5 million. HIV prevalence ranged from 4% to 9.6%, with approximately 259,000 persons receiving HIV care (DHIS2, 2020; Spectrum, 2021). Cellular coverage has expanded throughout the country, although pockets of poor coverage persist. Personal phone ownership remains generally low among clients accessing supported health facilities. Facility-based phones were utilized to deliver P-PSC services to people who did not own a personal phone.

### *Organization of psychosocial counseling before the COVID-19 pandemic*

Before 2020, 10 psychosocial counselors were each assigned to a high-volume health facility where they provided in-person counseling services. Psychosocial counselors assisted any person accessing services at these facilities, with clients accessing HIV services as the primary target. All psychosocial counselors held a diploma or degree in social work or psychosocial counseling from recognized training programs in Malawi. The psychosocial counselor training program lasted over four 20-week semesters. It included both didactic lectures covering various topics, including counseling theories and practice, ethics, counseling PLHIV, child and adolescent

counseling, marital and family counseling, substance abuse counseling and trauma counseling. The program also included clinical practical rotations. Upon being hired by Tingathe, the psychosocial counselors received didactic and practical orientation specific to care for PLHIV, including counseling to support ART adherence, HIV disclosure, coping with a new HIV diagnosis, intentional disclosure to children, and survivors of intimate partner violence. The mental health program lead (a mental health specialist) and district team leads provided technical and operational supervision.

Clients at all Tingathe-supported facilities were eligible for referral. Reasons for referral included treatment interruption, high viral load or ART adherence concerns, new HIV diagnosis, intimate partner violence, advanced treatment, disclosure support needs and mental health concerns, including depression, suicidality and alcohol/substance abuse. Tingathe-supported facilities without a resident psychosocial counselor would refer clients in need of psychosocial counseling services to the nearest facility with a psychosocial counselor, but very few of these referred clients actually traveled to receive psychosocial counseling services. Occasionally psychosocial counselors would travel to other Tingathe-supported facilities if a consultation was requested. However, this was logistically challenging and expensive, so such consultations were infrequent. Depending on the client's need, the psychosocial counselors provided supportive therapy, cognitive behavioral therapy, problem-solving therapy, psychoeducation or psychological first aid.

### P-PSC intervention

The P-PSC intervention was built on the existing psychosocial counseling infrastructure, and it transitioned from in-person counseling services to phone-based referral and counseling. See Figure 1. P-PSC implementation began concurrently during the first phase of national COVID restrictions in Malawi (June 2020) for clients from all 120 supported health facilities.

Due to COVID mitigation strategies, existing psychosocial counselors and facility staff were oriented to phone-based service delivery virtually. Virtual orientation utilized a combination of WhatsApp messages, voice notes and virtual meetings led by Tingathe's psychosocial program lead. Voice notes allowed for an explanation of new implementation strategies and new referral processes in real time to all sites when movement was limited, while follow-up virtual meetings provided time for clarification and discussion. Referral criteria and processes, logistics of space and privacy for counseling, confidentiality and linkage to site-based clinical teams for management of emergent needs were reviewed. Weekly meetings and a WhatsApp group were utilized to support initial technical, logistical and clinical challenges during the early implementation of P-PSC. Peer and supervisor feedback on support and management of clients with complex needs was prioritized during weekly meetings as well as didactic lectures as continuing education on relevant diagnostic and therapeutic strategies once initial referral systems and processes were well established.

All health facilities and psychosocial counselors were provided cell phones, ensuring access for clients from each facility regardless of phone ownership. During counseling sessions, the facility site supervisor handed the phone to the client, who spoke with the P-PSC counselor in a private space. These private spaces were places typically used for sensitive discussions, either a room or at times outside spaces where they could talk freely to the counselor on the phone.

During routine clinical encounters, eligible clients were offered P-PSC services by the facility staff. Referral criteria remained the same as before P-PSC services. New WhatsApp groups were created to manage referrals from all 120 facilities to the pool of psychosocial counselors. WhatsApp groups included a lay healthcare worker (HCW) from each facility, all psychosocial counselors and supervisory and administrative support (district and program leads, psychosocial program lead and a program assistant). When a person needing P-PSC services was identified, the lay HCW would send a message to their WhatsApp group with the facility's name, general reason for referral and facility phone number to call – no patient's personal identifying information was included. Psychosocial counselors monitored the WhatsApp groups throughout the

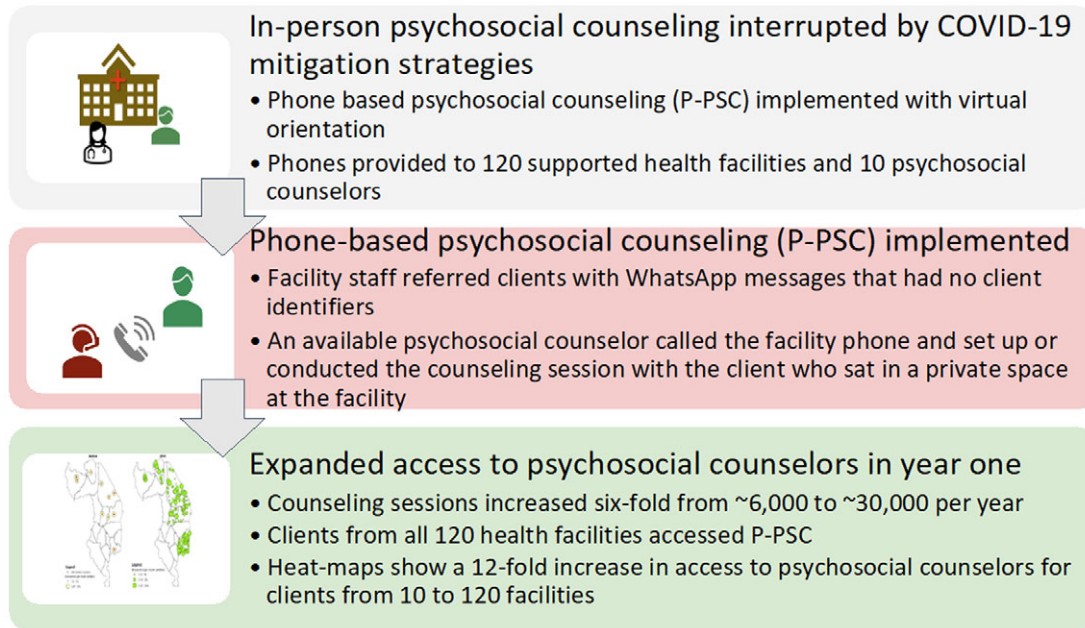

**Figure 1.** Implementation of phone-based psychosocial counseling (P-PSC): P-PSC increased access to psychosocial counseling in routine implementation of HIV care in Malawi.

day and attended to the consults in the order of receipt and acuity. The psychosocial counselor would phone the facility and talk directly to the referred person. The counseling session was either conducted immediately using the facility phone or was scheduled for a future time using the client's personal phone, based on client preference. For clients with emergent or complex needs, psychosocial counselors collaborated with the clinical provider at the client's facility to address immediate concerns. Consultation with the district mental health officers was made for further escalation of care as needed. At the initiation of P-PSC services, while everyone learned the new referral process, a program staff member co-monitored the WhatsApp groups to help assign referrals. During P-PSC implementation, demand for psychosocial counseling increased, so additional psychosocial counselors were recruited. On average, 13 psychosocial counselors daily provided P-PSC services.

### Program records data

Cohort characteristics, acceptability, feasibility and impact on psychosocial counseling utilization were examined utilizing routine program data. The program data included (1) pre-intervention in-person psychosocial counselor program reports submitted monthly with de-identified aggregated summaries of people counseled for 21 months (September 2018 to May 2020) and (2) post-intervention P-PSC Excel-based call logs submitted weekly by psychosocial counselors with de-identified patient information for 12 months (June 2020 to May 2021).

### Cohort characteristics

We examined the following cohort characteristics: (1) age, (2) sex, (3) reason for referral, (4) length of counseling session, (5) type of P-PSC session (individual, couple, family and group).

### Acceptability of P-PSC services

The acceptability of P-PSC was assessed at both the HCW and client level. At the HCW level, we reviewed the cadres of HCWs who made referrals to P-PSC. At the client level, we compared the total number of referred clients who completed P-PSC sessions to those who did not.

### Feasibility of P-PSC services and impact on access to psychosocial counseling services

The feasibility of implementation and impact of P-PSC on access to psychosocial counseling services was measured by examining the total number of P-PSC encounters per month, per day (average) and per counselor. The impact of P-PSC on the utilization of psychosocial counseling services was examined by evaluating the geographic coverage of P-PSC services. In either period (pre or post), geographic coverage was defined as the total number of facilities with a psychosocial encounter over the total number of Tingathe-supported health facilities. Impact analysis on the utilization of psychosocial counseling services used aggregate pre-intervention data from monthly program reports. These reports were compiled from the psychosocial counselors' paper records, providing a summary of services offered, including (1) the number of encounters by each psychosocial counselor, (2) reasons for referral, (3) age, (4) sex, (5) referred person's facility and (6) type of session from September 2018 to May 2020. The program did not collect individual-level data before P-PSC, so comparing between

pre- and post-intervention periods using individual-level data was not possible.

Additionally, details of post-intervention P-PSC counseling encounters allowed a description of the implementation. Data examined to describe P-PSC implementation included (1) psychosocial counselor's main facility, (2) date of the session, (3) referred person's health facility, (4) number of encounters by each psychosocial counselor, (5) method of counseling (phone or in-person), (6) encounter duration, (7) location of counseling, (8) personal or clinic's phone used and (9) type of visit (new or follow-up).

### Statistical analysis

Data were analyzed using univariable approaches and regressions. In univariable analysis, frequencies and proportions were used to summarize psychosocial counselor encounters; in the post-intervention period, these were disaggregated by sex, age and referral reason. We also compared the average number of encounters per counselor per month before and during the intervention. In additional subgroup analyses, we looked at referral reasons by age group and sex. In regressions, we evaluated the effect of P-PSC on psychosocial counselor coverage using single-group interrupted time series analyses (Kontopantelis et al., 2015). We were unable to use multiple-group interrupted time series analyses with control sites to check for historical threats because the intervention was implemented at the same time at all the sites supported by Baylor Foundation Malawi (Linden, 2017). Nonetheless, estimates from single-group interrupted time series analyses are still valid estimates for causal inference because the pre-post comparison within a single population in this design limits both selection bias and confounding, which often occur because of between-group differences (Bernal et al., 2017; Lopez Bernal et al., 2018).

We estimated the following model using ordinary least-squares regression:

$$Y_t = \beta_0 + \beta_1 T_t + \beta_2 X_t + \beta_3 X_t T_t \qquad (1)$$

where:

$Y_t$ is the average number of psychosocial counselor encounters in week $t$.

$T_t$ is the time since the study began. Its coefficient, $\beta_1$, represents the rate of psychosocial counselor coverage pre-intervention.

$X_t$ is a binary variable representing the study periods. Thus, $X_t = 0$ pre-intervention and $X_t = 1$ post-intervention. Therefore, $\beta_2$ represents the change in the number of psychosocial counselor encounters in the first week of intervention (immediate effect).

$X_t T_t$ is an interaction of time and intervention, with $\beta_3$ as the difference in the rate of psychosocial counselor coverage in the pre- and post-intervention periods.

$\beta_0$ is the number of psychosocial counselor encounters at the beginning of the study.

The model in Equation (1) was implemented in Stata 14.2, and all statistical tests were two-sided. Results were significant if the $p$ was <0.05.

### Autocorrelation

The model in Equation (1) was checked for autocorrelation (serial correlation), and the series was correlated. Autocorrelation refers to

the correlation between a variable and its previous values. It may lead to an inflated type 1 error (finding an effect where none exists) if positive, or an inflated type 2 error (not finding an effect where one exists) if negative. For the current study, autocorrelation was positive and up to lag 3 ($\chi^2$ (1) = 2.78, $p$ = 0.09). We accounted for the autocorrelation in regression analysis by incorporating three lagged values of the dependent variable as explanatory variables and Newey-West standard errors in the regression analysis. The Newey-West standard errors can handle both the problem of autocorrelation and heteroscedasticity (Newey and West, 1986). To verify that we had accounted for the correct autocorrelation structure, we applied the Cumby-Huizinga test for autocorrelation after running the regressions (Baum and Schaffer, 2013).

### Robustness of the analysis

We evaluated the primary assumption behind the interrupted time series analysis design and the model in Equation 1 by evaluating before the actual date of the intervention. Interrupted time series analysis assumes that the trend in the dependent variable (psychosocial counselor coverage) could have continued uninterrupted if the intervention had not been implemented (Bernal et al., 2017). Therefore, if this assumption holds, we should not find interruptions to the trend of psychosocial counselor coverage at any point before the true date of the intervention (Ukert et al., 2017). This assumption was checked in weeks 7 and 14 pre-intervention to allow enough time for the pre-intervention trend to form.

### Results

### Cohort characteristics

Table 1 shows the characteristics of people referred to P-PSC: more than half (63%) were female, 25% were youth (10–24 y), and 9% were children (<10 y). Referrals were for new HIV diagnosis (32%), adherence concerns (32%), treatment interruption (21%) and support for mental health needs, social needs, disclosure, abuse or other reasons (20%) (Figure 2). Referral reasons were significantly different between females and males ($X^2$ (7) = 266, $p$ < .001), with females more likely to be referred for mental health concerns and abuse. Males were more likely to be referred for a new HIV diagnosis, treatment interruption, social support and disclosure support. The proportion referred for adherence support was the same for both females and males. Children and adolescents (0–19 y) and adults over 35 y were most often seen for adherence support, while 20–34-y-olds were most often referred due to a new HIV diagnosis. Children and adolescents were most often referred for disclosure support (10–19 y: 63.9%; 1–9 y: 27.6%), $X^2$ (42) = 4,100, $p$ < .0001.

### Acceptability of the P-PSC intervention

The intervention was highly acceptable at the HCW level with lay HCW making 91% referrals, clinical staff 5% and self-referral 3%. The number of counseling sessions provided to clients increased from about 6,000 over 12 months before the intervention to 31,642 during 12 months of P-PSC intervention, representing more than a fivefold increase. Nearly every referred client (99.4%) received counseling during P-PSC implementation. Almost all counseling sessions (91%) took place at the clinic, while 7% were conducted over personal phones, offering flexibility to clients and increasing access. In a very small number of cases, less than 1% of clients could

**Table 1.** Summary of psychosocial encounters post-initiation of P-PSC (N = 31,128)

| Variable | Frequency | Percentage |
| --- | --- | --- |
| Sex | | |
| Females | 19,837 | 63 |
| Males | 11,781 | 37 |
| Missing data | 24 | 0 |
| Age (years) | | |
| <1 | 206 | 1 |
| 1–9 | 2,546 | 8 |
| 10–19 | 4,131 | 13 |
| 20–24 | 3,650 | 12 |
| 25–34 | 9,062 | 29 |
| 35–44 | 7,224 | 23 |
| 45+ | 4,410 | 14 |
| Missing data | 413 | 1 |
| Length of counseling session | | |
| <15 min | | 46 |
| 15–29 min | | 46 |
| 30–45 min | | 5 |
| 46+ min | | <2 |
| Unable to reach | | <1 |
| Missing data | | 1 |
| Type of P-PSC session | | |
| Individual | | 86 |
| Family | | 7 |
| Group | | 1 |
| Unable to reach | | <1 |
| Missing data | | 5 |
| New referral or follow-up visit | | |
| New referral | | 90 |
| Follow-up | | 8 |
| Missing data | | 2 |

not be reached, and data was missing for 2%. Despite only 45% of clients owning a phone, phone-based counseling sessions were accepted by clients with 82% of counseling sessions occurring on the phone, 15% in person, less than 1% unable to be reached and 3% missing data.

### Feasibility of P-PSC and impact on access to psychosocial counseling services

The P-PSC service was feasible with 31,642 P-PSC encounters in year 1, and a significant increase in the total number of P-PSC encounters per month, per counselor and per day (average) compared to the pre-P-PSC period (Table 2). A significant increase in both men and women counseled was also seen (Table 2). Regression results show that P-PSC had a positive impact as the number of counseling sessions increased (Figure 3). During the

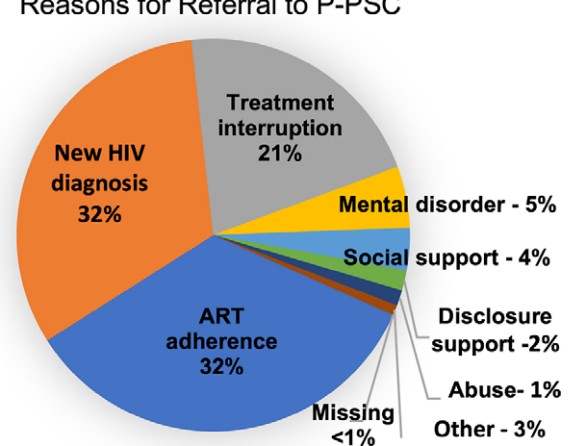

**Figure 2.** Reasons clients were referred to phone-based psychosocial counseling.

**Table 2.** Number of psychosocial counseling encounters pre- and post-intervention

| Variable | Pre-intervention | Post-intervention | p |
|---|---|---|---|
| Total (overall) | 10,504 | 31,642 | |
| Per month | 553 | 2,254 | <0.001 |
| Per counselor per month | 70 | 307 | <0.001 |
| Per day (average)[a] | 18 | 88 | <0.001 |
| Females per month[a] | 347 | 1,414 | <0.001 |
| Males per month[a] | 206 | 840 | <0.001 |

[a]The single-sample t-test was used. This compared whether the sample average during the intervention period was equal to the average (single value) in the pre-intervention period.

pre-intervention period, the average number of counseling sessions conducted by each counselor in the first month was 62, rising by an average of one session per counselor each month to a maximum average of 77 sessions per counselor per month. At the start of

P-PSC, a significant increase in counseling sessions occurred immediately, with each counselor conducting an average of 216 sessions per month starting at month 1 of P-PSC (95% CI = 82, 350, $p$ = 0.003). Throughout the 12 months of P-PSC reviewed, the number of encounters per month per counselor continued to increase, although the increase was insignificant.

The impact of P-PSC on geographic coverage and access to psychosocial counseling from all supported rural health facilities is shown in heat maps (Figure 4). The left of Figure 4 demonstrates the period prior to P-PSC with facilities marked yellow showing where psychosocial counselors physically sat daily and saw clients referred. The right side of Figure 4 demonstrates the P-PSC implementation period with facilities marked green showing clients counseled from all 120 supported health facilities. The size of circle depicts the relative number of clients counseled from each facility. Clients from every Tingathe-supported facility were now able to access P-PSC.

### Sensitivity analysis

In sensitivity analysis, the model in Equation (1) was robust when evaluated in pre-intervention weeks 7 and 14. In both weeks 7 and 14, there was no immediate effect and no significant change in the post-intervention trend (7th week: immediate change = −50 (95% CI = −113, 14), trend = 6.03 (95% CI = −2, 14); 14th week: immediate change = 3.61 (95% CI = −84, 91), trend = 10 (95% CI = −6, 27)). Therefore, the lack of evidence of interruptions in the pre-intervention period suggests that the interruptions observed in the 18th week of the study are likely due to the intervention.

### Discussion

We examined the acceptability, feasibility and impact of delivering P-PSC to people accessing HIV care at health facilities in Malawi. COVID-related restrictions presented an unexpected opportunity to expedite the adoption and implementation of P-PSC, enabling increased access to care for thousands of people in rural Malawi.

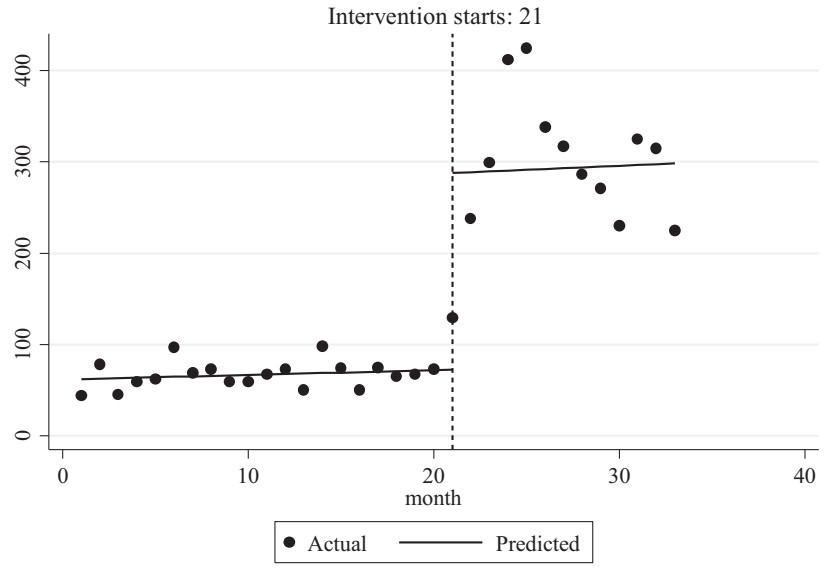

Regression with Newey-West standard errors - lag(3)

**Figure 3.** Interrupted time series analysis of impact of phone-based psychosocial counseling intervention on the number of counseling sessions.

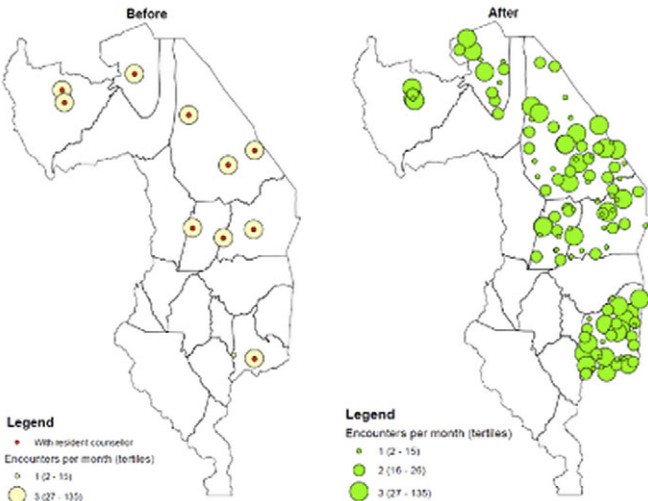

**Figure 4.** Heat maps demonstrating increase in the average number of counseling sessions per month at each facility before (left) and after (right) implementation of phone-based psychosocial counseling (P-PSC).

We demonstrated that phone-based delivery of psychosocial counseling services immediately increased the absolute number of people and facilities utilizing the services with increased geographic reach from 10 to all 120 supported facilities. Additionally, the implementation was sustained throughout the entire 1-y period examined. To our knowledge, this is one of the first published works on implementing P-PSC to remote facilities in sub-Saharan Africa. Our findings demonstrate that innovative and scalable interventions, such as P-PSC, can be successfully implemented in limited resource settings. They highlight acceptability, feasibility and impact of this promising intervention in addressing the established psychosocial health treatment gap that exists in limited resource settings like Malawi (WHO, 2017).

P-PSC allowed the program to expand the availability of counseling services twelvefold, from 10 to all 120 health facilities supported by Tingathe. Several factors likely contributed to this success. First, because services could now be offered via phone, counselors provided same-day psychosocial counseling to people from multiple facilities instead of just the facility where they were physically based. This rapidly increased the number of facilities and clients that could be served each day. Secondly, clients did not need to travel to access P-PSC services. Long distances to health facilities and the high cost of travel relative to income are known barriers to referral in Malawi (Pinto et al., 2013; Palk et al., 2020). Also, stigma toward mental illness is common (Crabb et al., 2012). P-PSC services at primary health facilities provided an option to access counseling without additional travel. This may have allowed more people to access counseling without explaining, taking time or paying for additional visits at far-away referral facilities. Finally, utilizing a facility phone aimed to give access to all clients, particularly those without phones and those who preferred not to use their personal phones. In our experience in rural catchment areas of supported health facilities, phone ownership is low overall with fewer women, children and people with disabilities having phone access. A phone dedicated to P-PSC services at each facility allowed anyone to access the service while at the health facility.

Our experience aligns well with existing literature describing the overall acceptability and feasibility of telehealth in African countries, as well as the increasingly common utilization of telehealth services during COVID-19-associated lockdowns (Daher et al., 2017; Adepoju, 2020). In Zimbabwe, during COVID-19 lockdowns, telemedicine was used to deliver obstetrics and gynecology consultations with high rates (94%) of patient satisfaction (Moyo and Madziyire, 2020). A study from a Zambian tertiary care facility's neurology clinic has demonstrated that telehealth was exceedingly acceptable to both clients (99%) and HCWs (100%) providing clinical consultation (Asukile et al., 2022). The DREAM program, supporting PLHIV for several sub-Saharan African countries, demonstrated that telemedicine consultation with a neurologist was also acceptable to PLHIV (Leone et al., 2018). Telepsychiatry was used in an outpatient psychiatry clinic in Kenya where focus group discussions demonstrated that participants overall had a positive experience, citing convenience and overall perceived effectiveness compared to in-person sessions (Kaigwa et al., 2022). In response to COVID-19 mitigation measures, the Zvandiri program, which caters to youth living with HIV in Zimbabwe, increased its use of virtual platforms for case management during home visits and extended support to its client base. This expansion included an increase in virtual support group offerings and enhancement in the preexisting Mobile Health program, now including mental health "check-ins" (Mawodzeke, 2020). In South Africa, MomConnect utilized phone messaging to share educational information with pregnant women as well as allowed interactive messaging through the help desk (Xiong et al., 2018). With COVID-19, the delivery of mental health services around the world largely transitioned to telehealth encounters. However, there was variability in implementation due to limitations in digital health platform access, patient preferences and individual medical needs (Li et al., 2022; Zangani et al., 2022).

The integration of P-PSC into existing service delivery also likely played a role in successful implementation. Sensitization to P-PSC services may have normalized counseling services for HCWs (both lay and clinical cadres) and clients at health facilities. As established members of facility healthcare teams and the surrounding catchment area community, lay HCWs were well positioned to improve mental health literacy for clients. Lay HCWs provided basic counseling information and support and then referred clients with additional needs to P-PSC. This collaboration aligns with reported opportunities to utilize the strengths of lay cadres to deliver psychosocial support services (Han et al., 2018).

Despite Malawi's known psychosocial health treatment gap, reported stigma associated with mental health needs, limited phone connectivity and limited human resources for health, P-PSC was quickly implemented at all 120 supported sites despite COVID-19 mitigation restrictions. Moreover, P-PSC services were sustained throughout an unprecedented year of healthcare service delivery interruptions due to the COVID-19 pandemic and continue to date (WHO, 2020; Thekkur et al., 2021; Zangani et al., 2022). Though the COVID-19 pandemic saw a dramatic increase in the use of telemedicine globally, published work on similar scaled use of telehealth for individual live-counseling sessions in sub-Saharan Africa is limited, making comparison difficult (Adepoju, 2020; Osei et al., 2021). This study provides additional data on the potential and successful use of telemedicine in addressing individual psychosocial support needs in resource-limited settings like Malawi. Additionally, while several studies have reported telehealth service delivery to clients from a single facility, we believe this is among the first reports of delivery of individualized live psychosocial counseling to clients from multiple, physically distanced health facilities simultaneously (Kaigwa et al., 2022).

## Limitations

This study had several limitations. Firstly, we could not estimate the absolute coverage of P-PSC. Secondly, the program implementation data did not capture all clients who met the criteria for referral to a psychosocial counselor, but just those ultimately referred to P-PSC. Therefore, the demand for P-PSC may have been higher than demonstrated here, and the number of eligible clients who declined the service was unknown. Notably, neither large unmet demand nor declined referrals were routinely observed or reported programmatically. The over 30,000 P-PSC encounters from all 120 supported health facilities suggest that the program provided an acceptable service for an unmet need. Additionally, although mobile phone penetration has improved in Malawi, some areas remain without stable connectivity, and intermittent electricity blackouts occur. Both issues may have affected successful referral and service delivery. Work is ongoing to monitor HCWs' and clients' experience by conducting programmatic exit interviews.

Furthermore, we could not compare the duration of counseling sessions before and after P-PSC implementation as the duration of sessions was not collected when psychosocial counseling was done only in person. With P-PSC, over 90% of visits were less than 30 min. This likely was shorter than in-person sessions before P-PSC, when very few clients were seen per day. This shortened counseling session aligned with published literature that phone-based counseling sessions were shorter than in-person ones. However, reassuringly, the literature saw no change in efficacy based on the duration of sessions with phone-based support (Irvine et al., 2020).

Finally, during this initial year of implementation, our programmatic data did not capture clients' outcomes. We focused on initial implementation with orientation and capacity-building for HCWs and psychosocial counselors while establishing referral systems. Outcomes for clients utilizing P-PSC services are now being followed to refine service delivery to meet clients' needs.

## Conclusions

Before the COVID-19 pandemic, telehealth was not an option for psychosocial counseling services in Malawi, but COVID-19 restrictions provided an opportunity to implement P-PSC. We found it to be both highly acceptable and feasible. Access to psychosocial counseling improved and remained sustainable over time. We are continuously working to improve the effectiveness of P-PSC through various approaches. We are recruiting more psychosocial counselors, strengthening their counseling skills and standardizing approaches, including routine screening and treatment for depression. Communication loops among psychosocial counselors, clients and clinical teams are being strengthened to provide holistic integrated service delivery. While HCWs returned to providing in-person care at the facility level soon after COVID-19 restrictions were lifted, P-PSC has continued to operate alongside in-person counseling. P-PSC service delivery to clients from all supported health facilities continues to date. This approach allows immediate, dramatic and sustained access to care for thousands of people, many of whom live far from psychosocial counselors. As an efficient, scalable model of providing counseling from trained psychosocial counselors in limited resource settings, P-PSC has the potential and further study is needed.

**List of abbreviations**

P-PSC      phone-based psychosocial counseling.
PLHIV     people living with HIV.
HCW       healthcare worker.

**Open peer review.** To view the open peer review materials for this article, please visit http://doi.org/10.1017/gmh.2023.84.

**Data availability statement.** The datasets analyzed in the current study are available from the corresponding author on reasonable request.

**Acknowledgements.** We are thankful to the Malawi Ministry of Health for support and partnership. We are grateful to the people living with HIV who participated in this implementation. We acknowledge the support of the Tingathe program team for their hard work and dedication.

**Author contribution.** C.M.C., S.M. and M.K. majorly contributed to manuscript writing. C.C., A.M., M.S., M.U., K.R.S., S.A. and P.N. oversaw program implementation. S.M. provided data analysis and interpretations. All authors contributed to manuscript review and final approval before submission.

**Financial support.** Baylor Foundation Tingathe is a USAID-funded program, Cooperative Agreement No. AID-674-A-16-000003.

**Competing interest.** The authors declare no competing interests exist.

**Ethics standard.** The retrospective program analysis was approved by the Malawi National Health Science Research Committee (protocol #688) and the Baylor College of Medicine Institutional Review Board.

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
