## [Reviewer Report]

Professor Judy Bass and Dr. Dixon Chibanda

Cambridge Prisms

Global Mental Health

29 March 2023

Dear Professor Bass and Dr. Chibanda:

I am writing to submit our manuscript entitled, “Phone-based psychosocial counseling for people living with HIV: feasibility, acceptability, and impact on uptake of psychosocial counseling services in Malawi” as a Global Mental Health research article.

As you are aware, there is a paucity of research on scalable psychosocial counseling options in low resource settings like Malawi. This research represents findings from a successful, real world, scale up of telephone-based psychosocial counseling in Malawi. We feel these findings will be of interest to readers of Global Mental Health.

We know of no conflicts of interest associated with this publication, and there has been no significant financial support for this work that could have influenced its outcomes. As Corresponding Author, I confirm that the manuscript has been read and approved for submission by all the named authors.

We declare that this manuscript is original, has not been published before and is not currently being considered for publication elsewhere. The retrospective program analysis has been approved by the Malawi National Health Science Research Committee (protocol #688) and the Baylor College of Medicine Institutional Review Board. The datasets analyzed during the current study are available on reasonable request.

Thank you for your consideration and we look forward to hearing if this article may be of interest to you.

Sincerely,

Carrie M Cox, MD

Corresponding Author

Assistant Professor of Pediatrics

Baylor College of Medicine Children’s Foundation Malawi

Baylor College of Medicine

cc12@bcm.edu

ccox@tingathe.org

---

## [Reviewer Report]

Thank you for the opportunity to review this manuscript. It summarises the accessibility and feasibility of a telehealth intervention that was implemented at scale for people living with HIV in Malawi during the Covid-19 pandemic. While the paper presents extremely exciting results about the implementation of such services at scale, it could be improved with the following additions and revisions.

The primary issue with this paper is that it does not detail anything about the type of counselling that was delivered, who delivered this counselling, how sessions unfolded, or what was discussed. The field of counselling for people living with HIV is highly developed, and there are many different kinds of counselling approaches - some that relate specifically to adherence to medication, stigma reduction, mental distress/common mental disorders, gender-based violence etc. These have been refined and adapted to Telehealth and mobile health settings in innovative ways, tailored to different populations like youth, women, and sex workers. A strong discussion of this field - at the intersection of mobile health, global mental health and HIV - is important from the outset of this paper, and should guide the way in which the intervention design is presented. The intervention should be analysed with specific comparison to a wider range of interventions in Sub-Saharan Africa that address the needs of this population - including, for example, MomConnect in South Africa, Khuluma and Insaka interventions by the SHM Foundation in South Africa, Zimbabwe, and Zambia, and Zvatinoda! Interventions in Zimbabwe.

Furthermore, there is little discussion of who delivered this counselling, how they were trained, and what their experiences of the program were. This intervention is comparable to other interventions that have used task shifting and lay health worker models to great effect, and others that train up professionals like social workers and nurses. Where does this intervention sit within this field, and why were such choices made? How did they impact the scalability of the model, the type of topic discussed, and the way that this intervention was integrated with the healthcare system.

This links to the third issue with the paper - the lack of detail on safeguarding, monitoring and referral processes. One of the biggest issues in Telehealth implementation is referral and monitoring. What was the process used here? What were the effects on acceptability and feasibility, especially at scale?

Fourth, the specificity of the Malawian setting also requires more detail in respect to four areas. (1) The phone - What does the phone mean in Malawi? Who has access? What kinds of issues surround phone use (connectivity, data access, infrastructure in rural areas, access to handsets, sharing of handsets, privacy)? What kinds of accessibility issues exist - related to disability, income and gender norms? How does this make the intervention more or less feasible/acceptable? (2) HIV and mental health care provision - what are the key needs mental health, physical health and adherence needs of populations living with HIV in Malawai? How does this compare to other countries in SSA where similar interventions have been delivered.

Fifth, the implementation process could be explained more clearly with a diagram, that could be paired with a similar diagram in the results section. More specific information of the acceptability of the referral process is needed, particularly with respect to the use of WhatsApp, as there are a range of privacy issues related to the software, including in message forwarding.

Sixth, there is little discussion of the different kinds of needs that might have been elided in the will to scale up the project. While there is some disaggregation of data, it is still unclear to whom the intervention is acceptable or feasible. For example, much literature suggests that young people need specific, tailored interventions to address their needs. Were these accounted for? To whom was this not acceptable and feasible?

Finally, I believe that some of the claims made in the discussion section should be tempered. For example, how could you know that the reason participants might have taken up the intervention about distance from healthcare facilities without having asked this of participants? There are other factors such as privacy and stigma that are also significant contributors to the success of phone based interventions.

Again, thank you for this opportunity to review this paper about an exciting intervention.

---

## [Reviewer Report]

This paper presents a clear description and analysis of phone counseling for people living with HIV in Malawi. While the findings regarding the impact of COVID-19 on use of phone counseling are obvious, the formal analysis nonetheless is in formative. A major limitation is the lack of detail on the background and supervision of the counselors. More detail is needed on their credentials and training. It would be important to know more about the duration and frequency of counseling sessions and whether the duration varied number of sessions. These details will be of use to anyone considering such services in the region.

---

## [Reviewer Report]

I would like to congratulate the authors on adding to the growing evidence of how best to integrate the everyday digital technologies in increasing the access to psychosocial counselling among HIV patients and improve their overall access to care.

There are some grammatical errors have been found in the manuscript, and individually, mentioning them is not possible. Kindly edit the whole manuscript accordingly. Also, please shorten the length of sentences, as it is creating a lot of confusion and misunderstanding of results. Listing a few below. A careful read would enhance the overall quality of the people.

1. Line 12 to 16. "Each counselor conducted an average of 216 ... the month before (CI: 82, 350, p-value=0.003)". This statement creates a slight confusion in the mind of the reader, due to its length, that the mentioned CI belongs to the previous month’s average, i.e., 82 - 350 is the CI of 77, which is not possible. Shorten the length of the statement, or divide it into two parts, so that it is easy to read and understand. Like in line 311-313, “At the start of ... month one of P-PSC ( ... )”.

2. Kindly rephrase and check for grammatical errors in the statement in Line 19 to 21.

3. Line 30, kindly check for grammatical errors. It should be “effects” instead of “affect”.

4. Line 36 to 39, kindly rephrase, Grammatical errors.

5. Line 45. Kindly check for grammatical errors. “Psychosocial health needs are needed” is a wrong statement.

6. Line 48. Kindly replace the word “requires” with “require”.

7. Line 60. Kindly change the word “Tingathe-supported lay”.

8. Line 63. Start the statement by putting “At” before “District and .... ”.

9. Line 247. Results having p-value equal to 0.05 are no longer considered as significant, or marginally significant. If any of your bivariable results are significant at 0.05 p-value, and you have used them in multivariable analysis or time-series analysis, kindly modify it. And, change the statement to “if p-value was <0.05.”

10. In Figure 2, Regression with Newey-West standard errors. After the intervention starts, the distribution of clients per counsellor per month has become very scattered, and their amplitude above the predicted line has gone beyond the 400 mark, whereas the rest remain in the mark of 210 - 330. Did you try transformation amongst the variables to make the distribution less skewed, as compared to what it is now?

11. Line 411, please insert a comma instead of a period after the word “Firstly”.

12. Line 412. The statement “Secondly, ... referred to P-PSC.” is grammatically wrong. There is no meaning to “identified to need P-PSC”. Kindly edit it.

13. Line 443. The statement “Leveraging ... implement P-PSC.” is grammatically wrong.

14. Line 448. The statement “Identification of ... client education.” is grammatically wrong.

14. Kindly change the tense of statements in the Limitations section to past tense.

---

## [Reviewer Report]

Dear Authors - thank you for this manuscript on integrated P-PSC with HIV care. I have read the manuscript along with the Reviewers and agree that while the paper has great merit, it seriously lacks in the description of P-PSC (what exactly was included, not included, etc.) and related the training content of for the phone counselors. This information is vital to understand how your findings complement and expand this field and--importantly--would allow other interested parties to better replicate your model.

My only addition to the reviews is to please use the word “sex” when you are referring to the biological attributes of the population (i.e., male vs female) and “gender” when you are referring to cultural constructs related to the expression of sex (i.e., masculine, feminine). Sex and gender are used interchangeably. I believe that you mean “sex” throughout.

One reviewer also had statistical queries that requires addressing.

Best, Jerome

---

## [Reviewer Report]

Thank you for your extensive and careful revisions to this article. It is much clearer and well improved. I especially commend the addition of figures, and the detail on training of HCWs. There are a number of smaller points that would further improve this article:

1. I am satisfied with the detail on the training that the HCWs received, but I think you like a discussion of the supervision or debriefing that the counsellors had. What happened in ‘weekly meetings’? How was quality counselling assured? How were difficult cases escalated? These are important elements to detail if you are claiming to have reached scale.

2. I am intrigued by the combination of tele-health technologies used, especially the combination of voice note, virtual meeting etc. This is a common, improvisational strategy that is not often detailed in academic write ups of interventions. Can you provide more detail on what this combination looked like and how it was devised?

3. There is a lack of discussion of privacy, data security and confidentiality concerns in the intervention. How were spaces private? How did you ensure that whatsapp groups were secure when it is possible to forward messages? You may not have answers to all of this, but it needs to be a bigger part of the discussion section at least.

4. There is little discussion of how the intervention was or could be sustained beyond the Covid period. You may not have this data, but discussion in the discussion or conclusions sections of whether this worked because of the exceptional moment, or whether it could or would be sustained beyond is important.

5. The discussion section would also be improved with deeper comparison to other examples of intervention that have delivered telehealth at scale during Covid. I am not convinced that this is the only intervention. If you want to make this claim, you need to define what you mean by scale. Comparative examples could involve Africaid Zvandiri’s interventions in Zimbabwe, or MomConnect in SA.

Thanks for this interesting article.

---

## [Reviewer Report]

Dear Authors: Thank you for this detailed revision of your manuscript. One of the Reviewers has a few minor points that I hope you will be amenable to addressing.

---

## [Reviewer Report]

Thank you for the comprehensive revisions to this article, it is strong and I look forward to seeing it in print.

Please just note that the reference you use to cite MomConnect’s work is dated to prior to Covid-19 - (Xiong et al. 2018)

---

## [Reviewer Report]

The paper written by Cox et al. is of great value. Thank you for the opportunity to review this manuscript.

- After revisions, the manuscript is in the best shape to be accepted for publication.